# Event timing in human vision: Modulating factors and independent functions

**Valtteri Arstila[1,2☉], Alexandra L. Georgescu[3☉], Henri Pesonen[4], Daniel Lunn[5], Valdas Noreika[6,7‡]\*, Christine M. Falter-Wagner[8‡]\***

**1** Department of Philosophy, University of Turku, Turku, Finland, **2** Turku Institute of Advanced Studies, University of Turku, Turku, Finland, **3** Department of Psychology, Institute of Psychiatry, Psychology and Neuroscience, King's College London, London, United Kingdom, **4** Department of Biostatistics, University of Oslo, Oslo, Norway, **5** Department of Statistics, University of Oxford, Oxford, United Kingdom, **6** Department of Biological and Experimental Psychology, School of Biological and Chemical Sciences, Queen Mary University of London, London, United Kingdom, **7** Department of Psychology and Speech-Language Pathology, University of Turku, Turku, Finland, **8** Department of Psychiatry, Medical Faculty, LMU Munich, Munich, Germany

☉ These authors contributed equally to this work.
‡ These authors are joint senior authors on this work.
\* v.noreika@qmul.ac.uk (VN); christine.falter@med.uni-muenchen.de (CMFW)

**Data Availability Statement:** Data publicly available on OSF (DOI 10.17605/OSF.IO/3KHDG).

**Funding:** We would like to thank the participants for volunteering their time and thoughts. This work

## Abstract

Essential for successful interaction with the environment is the human capacity to resolve events in time. Typical event timing paradigms are judgements of simultaneity (SJ) and of temporal order (TOJ). It remains unclear whether SJ and TOJ are based on the same underlying mechanism and whether there are fixed thresholds for resolution. The current study employed four visual event timing task versions: horizontal and vertical SJ and TOJ. Binary responses were analysed using multilevel binary regression modelling. Modulatory effects of potential explanatory variables on event timing perception were investigated: (1) Individual factors (sex and age), (2) temporal factors (SOA, trial number, order of experiment, order of stimuli orientation, time of day) and (3) spatial factors (left or right stimulus first, top or bottom stimulus first, horizontal vs. vertical orientation). The current study directly compares for the first time, performance on SJ and TOJ tasks using the same paradigm and presents evidence that a variety of factors and their interactions selectively modulate event timing functions in humans, explaining the variance found in previous studies. We conclude that SJ and TOJ are partially independent functions, because they are modulated differently by individual and contextual variables.

## 1. Introduction

### 1.1 Event timing abilities: Perception of simultaneity and temporal order

As humans, we can achieve a coherent representation of our external world through information processing. One essential prerequisite for successfully processing information is the ability to place perceptual events relative to each other in time. This allows us to understand relationships in the world and, consequently, to detect structures and patterns around us. Therefore,

was funded under the Volkswagen Foundation grant I/82 894 awarded to VA, VN and CFW.

**Competing interests:** The authors have declared that no competing interests exist.

advancing our understanding of the mechanisms behind this ability as well as the temporal limits of the sensory mechanisms involved is central for understanding how we perceive and interact with a world that is fundamentally dynamic. The two common methods to measure such event timing abilities are the simultaneity judgement task (SJ task) and the temporal order judgment task (TOJ task). In both tasks, participants receive sensory stimulation with two stimuli, separated by a certain stimulus onset asynchrony (SOA). In the SJ task, participants have to judge whether the presented stimuli are simultaneous or not [1–4]. In the TOJ task, they have to judge which of two presented stimuli appeared first/last [5–8]. In both tasks trial-by-trial data can be summarized via meaningful model parameters when an appropriate observer model is fitted. The judgement proportion of SJ as a function of the SOA, exhibits a curve that can be fit to a Gaussian frequency function. Similarly, the judgement proportion of TOJ as a function of the SOA typically exhibits a sigmoidal curve that can be fit to a cumulative Gaussian function.

### 1.2 Diversity of findings on event timing thresholds

Interestingly, research using such tasks generally shows a large variability in findings related to event timing judgments. In SJ tasks, the simultaneity threshold, defined as the SOA at which participants' respond half the time that the two asynchronous stimuli are simultaneous, varies depending on the sensory modality. The best temporal resolution (i.e., the lowest threshold of detection of non-simultaneity) is observed in the auditory system, where two clicks presented to different ears only 2–3 ms apart are detected as non-simultaneous [9]. The visual modality, on the other hand, has the lowest temporal resolution with thresholds of some tens of milliseconds (i.e. 50–60 ms, [10,11]. The tactile system is somewhere between them, as its simultaneity threshold is often found to be approximately 20 milliseconds approximately 20–30 milliseconds [12,13]. For multimodal stimuli (i.e. stimuli in more than one modality, like a visual flash and an auditory beep), the simultaneity threshold can be more than 100 milliseconds [2,14,15].

The temporal order threshold is usually defined as the SOA at which participants' responses are correct 75 per cent of time [5,8]. Here, the threshold is around 20 ms for trained participants [5,16]; and up to 60 milliseconds for untrained participants [17] at least when auditory stimuli are used. As opposed to the simultaneity threshold, the temporal order threshold has been found to be rather consistent across sensory modalities as well as with multimodal stimuli, and it has been suggested to result from some kind of central time-organizing system that is independent of peripheral sensory mechanisms [5,15,16,18–23]. However, huge variability in TOJ thresholds has also been observed, depending on the physical properties of the stimuli (for a review, see Wittmann, 2011a) and even different task version [19,24,25]. In the case of visual stimuli, the direction of apparent movement, brought about by the stimuli, allows participants to infer which stimulus appeared first [26,27]. In the case of auditory stimuli, Mills and Rollman [28], for example, found TOJ thresholds of about 60 ms for click stimuli. Others found very low thresholds of 10 ms for tones of different frequencies [29]. Interestingly, participants can use several different cues ranging from the location of the stimuli [30] to the loudness [31] and the dominant pitch [32].

### 1.3 A debate on mechanisms

One important debate that has arisen from the diversity of findings above is whether the perception of successiveness (i.e. non-simultaneity) is merely a necessary condition for the perception of temporal order, or whether it is also a sufficient condition. At the heart of the matter is the question whether the two types of event timing judgements, SJ and TOJ, reflect

the same internal events or not. In theory, if two stimuli (x and y) are shown, a detector comparing the arrival times of sensory volleys related to them ($T_X$, $T_y$) could provide both SJ and TOJ. If $T_X$-$T_y$ is close to 0 ms (or below a certain threshold), they are judged simultaneous; if $T_X$-$T_y$ differs from 0ms (or is above a certain threshold), the order is determined based on the positive versus negative value of the equation. In this model, the perception of successiveness (i.e. non-simultaneity) is a necessary and sufficient condition for the correct perception of temporal order [33] (for review see [34]). Several well-known (and previously popular) positions—such as the perceptual latency model [35,36], the attention-switching model [37,38] and the triggered moment view [22,39]—have subscribed to this idea. According to the triggered moment view [39], for example, the first sensory volley triggers an atemporal system state and two stimuli are considered simultaneous, or co-temporal, only if the second sensory volley arrives during this system state.

On the other hand, SJ and TOJ do have different requirements: SJ requires information about the deviation from the 0ms whereas TOJ requires the direction of the deviation. Accordingly, it is not obvious that both requirements are best satisfied with one mechanism. This leads to the idea that the perception of successiveness/simultaneity is a necessary yet not sufficient condition for the perception of temporal order. Hirsh and Sherrick [5] proposed this view based on the facts that the SJ threshold for auditory modality is much lower than the TOJ threshold and that only the SJ thresholds appear to depend on the modality of the stimuli. Moreover, recent evidence has been accumulating showing that some psychophysical effects dominantly or specifically seem to occur only in TOJ but not in SJ (e.g. the crossed-arm deficit observed in tactile TOJ tasks, [33]) and it might be that perceptual processes needed for SJ are included in those for TOJ, but that TOJ might require additional perceptual processes to SJ [5,21,40–43]. This view has been developed further in the recent decades by describing the nature of the two mechanisms underlying both SJ and TOJ, their interrelation and how different factors influence them differently [33,44,45].

## 1.4 Methodological differences in event timing research

In sum, it is worth noting that despite the interest the topic has received, findings remain inconclusive. This might be due to the fact that the debate concerning SJ and TOJ often does not consider the differences in research tasks. One crucial difference is between the studies using unimodal stimuli and those using multimodal stimuli. Comparing the findings of these studies is problematic since SJ thresholds of unimodal stimuli are thought to result from modality specific mechanisms—the found thresholds are of different magnitude with those from multimodal stimulation leading to the highest thresholds [2,14,15]. Moreover, some of the studies use a ternary response task, rather than a SJ and/or TOJ task. In this task, SJ and TOJ are combined as one task, meaning that the participant is required to say whether the stimuli were presented simultaneously or not, and in the latter case, to identify which one of the stimuli was presented first (e.g. [34,45–47]). While using such a task is arguably not problematic if SJ and TOJ result from a single mechanism, the findings obtained in this way have been used against the view of a single mechanism behind SJ and TOJ [46,47]. Indeed, very few studies investigate event timing abilities by using the same stimuli across SJ task and TOJ task [43,45,48]. A final, often overlooked, difference in the studies is that while the judgments are most often made based on the onset of stimuli, on some much-discussed studies they are made based on offset of stimuli (e.g., [34,38]. Yet, the auditory system, for instance, reacts differently to onsets and offsets, and this is also reflected on the generalization effects between SJ and TOJ tasks [49,50].

## 1.5 Aim of the present study

In short, there is a long history in experimental psychology of investigations on the human ability to judge temporal relations of events. However, advancements in understanding this ability and the cause of the large variability in judgements has been partly hindered by the fact that different phenomena (unimodal vs. multimodal tasks, onset vs. offset tasks) are possibly confound together in previous research paradigms. Therefore, the aim of the present study is to approach the question of whether SJ and TOJ are based on the same underlying function or not, by investigating whether they are modulated by the same contextual and individual factors in several visual event timing tasks that differ solely in terms of stimulus arrangements and are comparable across all other features. TOJ and SJ paradigms were employed in a unimodal visual context while controlling for the following factors: (1) Individual factors (sex and age of participants), (2) Temporal factors (SOA, trial number, order of task, order of stimuli orientation, time of day) and (3) Spatial factors (left or right stimulus first, top or bottom stimulus first, horizontal vs. vertical orientation). To date no study has been undertaken that combines this variety of different factors in a unimodal paradigm to test both SJ and TOJ.

We hypothesized that all included factors will have an effect on either SJ, TOJ or both: With respect to the factor Sex, we hypothesized lower TOJ thresholds in men than in women, based on previous research [15,20,29,51–58]. Nevertheless, to our knowledge, this has not been investigated in the visual domain or for SJ judgements. In concordance with previous findings on the influence of the factor Age, we hypothesize higher thresholds with increasing age [59] and TOJ [55,56]. We further hypothesize positive learning effects with increasing Trial Number, based on empirical data from TOJ tasks [49,50,60–65] and SJ tasks [49,50]. To our knowledge, no generalization of training effects for SJ and TOJ have been done in the visual domain, therefore our investigation of the factor Order of Task is exploratory. Only one other study has investigated the Orientation factor in an SJ task and did not find any significant differences between target orientations [44], therefore the effect of orientation or the order of stimuli orientation for SJ and TOJ will be exploratory. In concordance with findings by Lotze and colleagues [15] we hypothesized a modulatory effect of the factor Time of Day for SJ but not TOJ. The TopBottom factor has not been studied yet, to our knowledge, however, there is evidence from TOJ studies, that the probability of responding "left" is higher when the left bar is presented first than when the right one was presented first due to attentional effects [66].

## 2. Methods

### 2.1 Participants

A total number of 35 students from the University of Turku were recruited to take part in the current study. Data of one participant was not included in the analysis due to random response behaviour (performance up to 4 SD below the group mean). The final 13 male (mean age = 27.8, SD = 6.6, min = 22 years, max = 46 years) and 21 female participants (mean age = 22.5, SD = 2.0, min = 19 years, max = 27) had normal or corrected-to-normal vision and were right handed (with the exception of one who was ambidextrous, mean laterality quotient was 82). Participants received credits for their participation as specified in their study programmes. Informed consent was obtained from all participants prior to testing and ethical approval was obtained from the University of Turku Ethics committee.

### 2.2 Apparatus

Stimuli were presented on an 18-inch desktop PC at a refresh rate of 120 Hz in a quiet standardized testing room. Participants viewed stimuli from a distance of 90 cm in an environment

of ambient light (90 lux), so as to reduce the impact of onscreen persistence. Inquisit 3 Software (2003) was used for experimental control and stimulus presentation.

## 2.3 Stimuli and procedure

Each participant performed four task versions: a horizontal simultaneity judgement task (HSJ), a vertical simultaneity judgement task (VSJ), a horizontal temporal order judgement (HTOJ) task, and a vertical temporal order judgement (VTOJ) task, which were conducted in counterbalanced order (i.e. either the two SJ or the two TOJ tasks were presented first in counterbalanced order with either the horizontal or the vertical task version being presented first in counterbalanced order). The HSJ task was exactly replicated from our previous study [67] and the VSJ version was adapted to vertical stimulus presentation. The HTOJ and VTOJ were exactly the same tasks only with different instruction (i.e. asking participants to attend to the temporal order of stimuli instead of their simultaneity). In each of the experimental versions participants were presented with trials starting with a white fixation cross that was presented centrally on a black background and remained on the screen throughout the trial (see Fig 1). Participants were instructed to fixate on the fixation cross and avoid eye blinks or movements during stimulus presentation. 500 ms after fixation cross onset two vertical bars (4.5 centimetres x 0.4 centimetre) subtending 4.1x0.4 degrees of visual angle with 11.1 degrees of visual angle between their centres were presented to the left and right (in the horizontal SJ and TOJ experiments) or above and below (in the vertical SJ and TOJ experiments) of the fixation cross. The bar stimuli were faded in incrementally within 5 frames (8.33ms per frame) from 5.2–53.7 lux (and kept at that brightness until the response), which prevent the phenomenon of apparent motion. Stimuli were either presented simultaneously or in one of 12 stimulus onset asynchrony (SOA) conditions defined by the monitor refresh rate (i.e. 0–12*8.33 ms) resulting in SOAs of 8.33, 16.67, 25, 33.33, 41.67, 50, 58.33, 66.67, 75, 83.33, 91.67, and 100 ms. The SOA conditions as well as the presentation of the first bar on the left or right (top or bottom respectively) were randomised on an experiment-wise basis.

In the simultaneity judgement experiments (HSJ and VSJ) participants were instructed to indicate whether the two bars appeared simultaneously or asynchronously by pressing one of two keys (4 vs. 6) on the keyboard number pad. In the temporal order judgement experiments (HTOJ and VTOJ) participants were instructed to indicate which bar appeared first by pressing one of two keys. In the HTOJ task they had to indicate whether the left bar or the right bar appeared first using the left arrow key and the right arrow key respectively. In the VTOJ task they had to indicate whether the top bar or the bottom bar appeared first using the up arrow key and the down arrow key respectively. In all tasks no feedback on performance was provided and the bars remained visible on the screen until key press. After each trial, confidence ratings were collected, which will be reported elsewhere.

Participants were shown five practice trials to get acquainted with the task requirements. The experimenter ascertained that the participants had understood the task instructions before commencing with the testing. Each task lasted for about 10–15 minutes and consisted of five blocks of 52 trials, resulting in 260 trials per experiment.

## 2.4 Statistical analysis

The binary responses ('simultaneous' vs. 'asynchronous' in HSJ and VSJ; 'left bar first' vs. 'right bar first' in HTOJ; 'top bar first' vs. 'bottom bar first' in VTOJ) were modelled using multilevel logistic regression [68–70], which accounts for within-subject correlation resulting from repeated measures carried out on individuals. In all of the models the $i$th response is of a

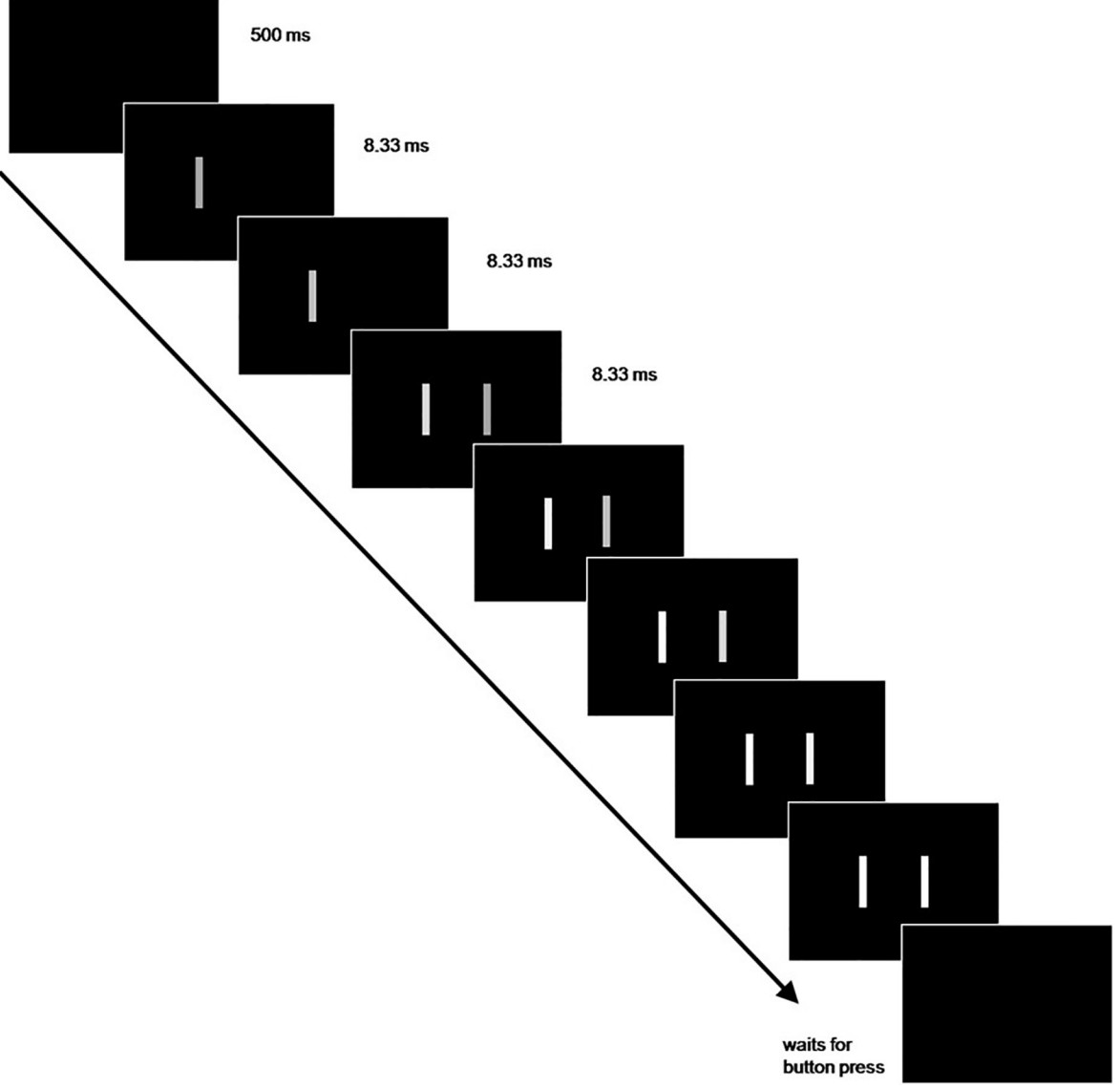

**Fig 1. Visual representation of the trial structure of the paradigm.**

general form

$$\mathrm{logit}(\mathrm{Pr}(y_i = 1)) = \log\left(\frac{\mathrm{Pr}(y_i = 1)}{\mathrm{Pr}(y_i = 0)}\right) = X_i\beta + Z_i B_{j[i]} \ , \ i = 1, \ldots, n, \ j = 1, \ \ldots, \ 34$$

where $y_i$ is the coded binary response, $X_i$ is a matrix of explanatory variables and $\beta$ are the fixed coefficients common to all subjects. In addition, $Z_i$ and $B_{j[i]}$ are the matrix of explanatory variables for the subject-level random modelling and the random coefficients, respectively. The notation $j[i]$, indicates which subject $j$ produced the $i$th response. Random coefficients are modelled as normally distributed

$$B \sim N(0, \ \Sigma),$$

where the covariance matrix $\Sigma$ is inferred along with the parameters $\beta$. In all models, subjects

were allowed to have their own base level of obtaining response $y_i = 1$, along with their own spatial context effect on the response. This is achieved by modelling respective coefficients as random. Models were fitted and the statistical significances of the parameters obtained using R software [71] and MASS package [72]. Results were further confirmed by checking against estimates from the MLwiN package [73] (Zhang, Parker, Charlton, Leckie, & Browne, 2016).

The binary responses were coded as 1 if the response was 'asynchronous' or 'left bar first' or 'top bar first', as appropriate. The explanatory variables included and tested were either concerning (1) individual factors, or (2) temporal or (3) spatial context. From the category of "individual factors", we included *Age* and *Sex* of the participants. From the category "temporal context", we included *SOA* (i.e. Stimulus Onset Asynchrony between the two bar stimuli), *TrialN* (i.e. the trial number which allowed testing the influence of a training effect within each of the experiments), *Order_Exp* (i.e. whether SJ or TOJ were conducted first), *Order_Orientation* (i.e. whether the horizontal or the vertical task version was conducted first) and *Daytime* (i.e. the time of day when participants conducted the tests). From the category of "spatial context", we have included *LeftRight* (i.e. whether the left or the right bar appeared first in any one trial in the horizontal task versions), *TopBottom* (i.e. whether the top or the bottom bar appeared first in any one trial in the vertical task versions) and *Orientation* (i.e. whether stimuli were presented in a horizontal or vertical display). Whereas the inclusion of the majority of these factors was planned, *Age*, *Sex* and *Daytime* were opportunistic variables, the results of which must be interpreted with caution.

The simultaneity judgement tasks (HSJ and VSJ) were analysed together. The variables *Orientation*, *LeftRight*, and *TopBottom* could not be entered into the same regression model because of mutual exclusivity. Hence, three models were fitted in order to determine the individual influence of each of these variables on simultaneity judgements. In the first model, *Orientation* (horizontal vs. vertical) was entered as a variable in order to determine the influence of horizontal versus vertical stimulus presentation on simultaneity judgements. The second model tested the horizontal task version alone with *LeftRight* entered as a variable into the model in order to determine whether it made a difference for simultaneity judgements when the left or the right bar was presented first. The third model tested the vertical task version alone with *TopBottom* entered as a variable into the model in order to determine whether it made a difference for simultaneity judgements when the top or the bottom bar was presented first. The equivalent procedure was adopted for the temporal order judgement experiments (HTOJ and VTOJ). In all SJ logistic regression analyses the dependent variable entered was response type ('synchronous' or 'asynchronous') per trial. The first logistic regression analysis of TOJ with *Orientation* entered as a variable used proportion of correct responses as the dependent variable, because in contrast to SJ, where proportion of 'asynchronous' responses served as the dependent variable, there was no unified response type in TOJ (proportion of 'left' responses in the LeftRight version of the task and proportion of 'top' responses in the TopBottom version of the task). However, the second analysis with *LeftRight* entered as a variable was modelled with proportion of 'left' responses as the dependent variable and the third analysis with *TopBottom* entered as a variable was modelled with proportion of 'top' responses as the dependent variable.

## 3. Results

The reported models include only factors, i.e. explanatory variables that had a significant effect on interpretational outcome, which are shown in the tables below. Factors that did not significantly contribute to a model are not in the output and not shown in the tables. Sometimes factors in the tables have p-values that are not significant, these factors are shown because they

**Table 1. Mean percentage of responses across tasks.**

| SOA (ms) | 0 | 8.33 | 16.66 | 25 | 33.33 | 41.66 | 50 | 58.33 | 66.66 | 75 | 83.33 | 91.66 | 100 |
|---|---|---|---|---|---|---|---|---|---|---|---|---|---|
| VSJ | 89.71 | 86.03 | 81.13 | 68.63 | 44.85 | 28.68 | 12.01 | 7.11 | 6.13 | 3.68 | 2.21 | 2.45 | 1.96 |
| | (12.15) | (16.50) | (15.67) | (16.29) | (25.38) | (23.32) | (14.53) | (13.16) | (12.19) | (8.00) | (5.91) | (5.24) | (3.59) |
| HSJ | 89.22 | 87.01 | 76.72 | 61.27 | 38.24 | 21.08 | 8.82 | 4.66 | 3.68 | 1.96 | 0.74 | 1.23 | 0.49 |
| | (12.23) | (12.68) | (19.11) | (26.02) | (25.55) | (25.48) | (14.50) | (8.51) | (5.87) | (4.61) | (2.40) | (4.65) | (1.99) |
| VTOJ | - - | 57.35 | 57.35 | 62.75 | 69.36 | 79.17 | 85.29 | 87.01 | 92.89 | 93.38 | 95.59 | 94.12 | 93.63 |
| | - - | (27.70) | (26.67) | (20.90) | (23.88) | (21.31) | (19.30) | (18.56) | (12.98) | (11.93) | (9.42) | (11.35) | (12.64) |
| HTOJ | - - | 56.13 | 56.62 | 65.93 | 74.26 | 86.52 | 90.44 | 93.63 | 94.61 | 95.10 | 96.32 | 93.87 | 97.55 |
| | - - | (21.65) | (27.26) | (24.39) | (23.51) | (15.86) | (14.74) | (12.80) | (9.32) | (8.57) | (6.63) | (12.14) | (8.70) |

Mean percentage (SD) of 'simultaneous' responses in the vertical and horizontal simultaneity judgements tasks (VSJ, HSJ) and mean percentage (SD) of correct responses in the vertical and horizontal temporal order judgement tasks (VTOJ, HTOJ) across stimulus onset asynchronies (SOA). There are by definition no correct responses in the 0 ms condition for TOJ tasks.

contribute significantly within an interaction. Average responses (and *SD*) for each of the experiments are depicted in Table 1.

## 3.1 Simultaneity judgement experiments

The explanatory variables that had a significant effect on interpretational outcome are listed in Table 2. In the SJ Orientation model, by far the strongest influence on simultaneity judgements was exerted by *SOA*, which shows that the longer the SOA, the less likely participants were to respond 'simultaneous' (see Coefficients depicted in Table 2). The larger the SOA the more likely participants were to respond 'asynchronous', as for each additional frame (8.33 ms) the

**Table 2. Binary regression analyses of simultaneity judgements (SJ).**

| Model | Variable | Coefficient | *SE* | Odds-ratio | *p*-value |
|---|---|---|---|---|---|
| Orientation | (Intercept)[R] | -3.714 | .164 | .024 | < .001 |
| | SOA[1] | .777 | .025 | 2.175[1] | < .001 |
| | TrialN[2] | -.004 | .001 | 1.837[2] | < .001 |
| | Gender (male) | 1.134 | .203 | 3.110 | < .001 |
| | Orientation[R] (horizontal) | -.299 | .299 | .742 | .040 |
| | Order_Exp (TOJ first) | .850 | .850 | 2.340 | < .001 |
| | Gender X SOA | -.143 | .036 | .867 | < .001 |
| TopBottom | (Intercept)[R] | -4.051 | .313 | .017 | < .001 |
| | SOA[1] | 0.771 | .036 | 2.162 | < .001 |
| | TrialN[2] | -.003 | .001 | 1.588 | .024 |
| | Gender (male) | .891 | .374 | 2.437 | .024 |
| | TopBottom[R] (bottom first) | .318 | .123 | 1.375 | .010 |
| | Order_Exp (TOJ first) | .615 | .260 | 1.850 | .024 |
| | Gender X SOA | -.116 | .054 | .890 | .030 |

Coefficients, standard errors (*SE*), odds-ratios, and *p*-values are only reported for significant variables. The direction of significant effects in terms of conditions with higher likelyhood of responding 'simultaneous' is given in brackets behind variables. Model LeftRight is not presented, because there was no significant effect of variable LeftRight.

[R] random effects (as opposed to fixed effects)

[1] SOA is a scalar variable and the odds ratio was calculated per unit of change (i.e. per refresh frame of 8.33 ms)

[2] TrialN is a scalar variable and the odd ratio was calculated between the first and the last trial of the experiment

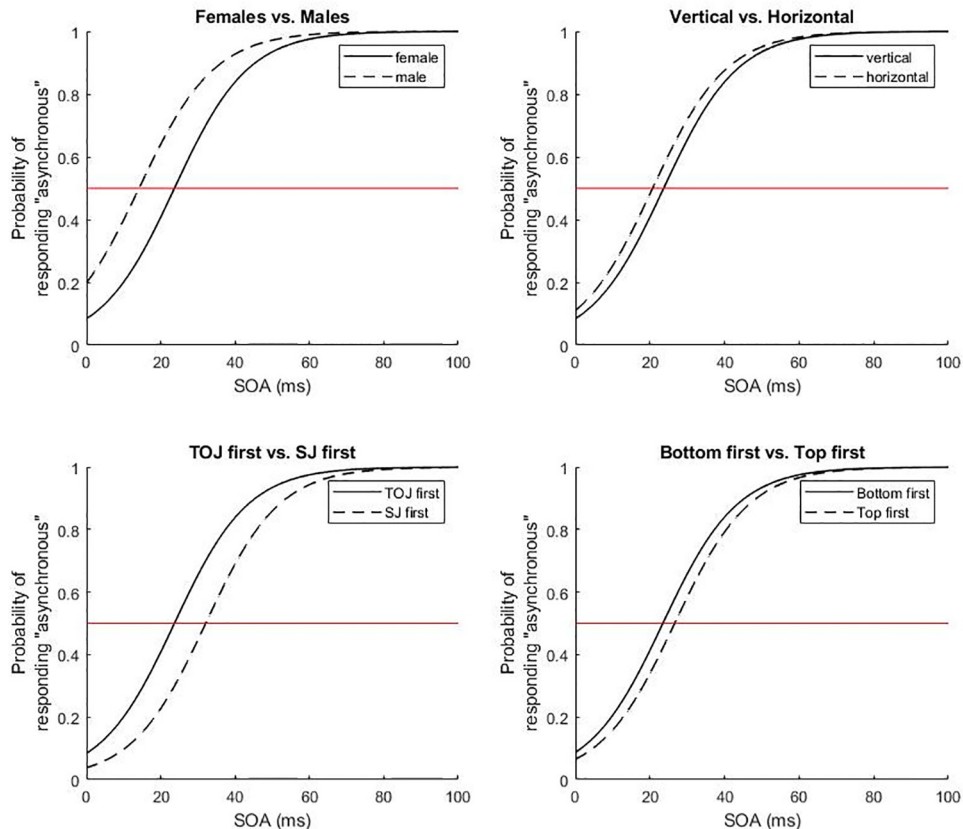

**Fig 2. The four plots represent illustration examples for factor combinations in the Orientation model (upper two and lower left plots) and the TopBottom model (lower right plot).** Probabilities of responding 'asynchronous' over stimulus onset asynchronies (SOA) in the SJ task for (upper left) females (solid line) versus males (dashedline), (upper right) vertical (solid line) versus horizontal bar depiction (dashed line), (lower left) TOJ first (solid line) versus SJ first (dashed line), and (lower right) presentation of bottom bar first (solid line) versus top bar first (dashed line).

likelihood of responding 'asynchronous' doubled (see odds ratios depicted in Table 2). Note that the effect of *SOA* was evaluated for two adjacent SOA conditions (e.g. 8.33 ms and 16.67 ms), hence, the effect across the range of SOAs is very large. The variable *TrialN* also influenced simultaneity judgements significantly. Participants were less likely (odds ratio = 0.538) to respond 'asynchronous' at the end rather than at the beginning of the experiments.

Furthermore, we found a significant effect of *Sex* and an interaction of *Sex* and *SOA*. All other variables kept the same, male participants were more likely (odds ratio = 2.696) to have responded 'asynchronous' than female participants (see also Fig 2). Moreover, the interaction between *Sex* and *SOA* shows that *SOA* influenced *Sex* differently with respect to their 'asynchronous' judgements. This interaction also indicates that the male response bias towards responding 'asynchronous' was largest at intermediate SOAs.

The variable *Orientation* exerted a significant influence on the amount of SJ in that the vertical presentation of the two bar stimuli (i.e. under and above the centre of the screen) rendered participants less likely (odds ratio = .742) to respond 'asynchronous' than the horizontal presentation of the two bar stimuli (i.e. to the left and right of the centre of the screen) suggesting a response bias for asynchrony for horizontal stimulus presentation. A significant main effect for *Order_Exp* was found. All other variables being equal, a participant who performed TOJ before SJ was more likely (odds ratio = 2.34) to have responded 'asynchronous' than a

**Table 3. Thresholds (ms) of simultaneity judgements.**

| Model | Variables | SJ first | TOJ first |
|---|---|---|---|
| Orientation | HSJ | 36.8 (female) | 28.4 (female) |
| | | 31.4 (male) | 21.1 (male) |
| | VSJ | 39.8 (female) | 31.4 (female) |
| | | 35.0 (male) | 24.7 (male) |
| TopBottom | Top bar first | 40.4 (female) | 34.3 (female) |
| | | 37.2 (male) | 29.9 (male) |
| | Bottom bar first | 37.3 (female) | 31.1 (female) |
| | | 33.4 (male) | 26.2 (male) |

participant who performed SJ before TOJ. Hence, performing a TOJ task beforehand increases the resolution in a subsequent SJ task.

These results indicate that simultaneity thresholds varied with several factors. The dependency of simultaneity thresholds on *Sex*, *TopBottom*, and *Orientation* can be seen in Table 3.

The logistic regression analysis of the second model with LeftRight entered as a variable showed no significant difference for the left bar being presented first in comparison to the right bar being presented first. The third model with *TopBottom* entered as a variable showed a significant difference between the top bar being presented first and the bottom bar being presented first. All else being equal, participants were more likely (odds ratio = 1.375) to respond 'asynchronous' when the bottom bar was presented first, than when the top bar was presented first, suggesting that the response bias towards "asynchronous" increased if the bars were presented in the order bottom-top compared to when they were presented in the order top-bottom.

## 3.2 Temporal order judgement experiments

The Orientation model showed that *SOA* exerted a statistically significant influence on TOJ (see significant explanatory variables in Table 4). The longer the SOA the more likely participants were to correctly indicate the temporal order of the bars. For each additional frame (8.33 ms) participants were more likely (odds ratio = 1.049) to respond correctly. Although the variables *Sex* and *Orientation* did not reach significance, there were significant interactions between *SOA* and *Sex*, indicating that the female participants were more accurate than male

**Table 4. Binary regression analysis of temporal order judgements (TOJ).**

| Model | Variable | Coefficient | SE | Odds-ratio | p-value |
|---|---|---|---|---|---|
| Orientation | (Intercept)[R] | -1.357 | .127 | .257 | < .001 |
| | SOA[1] | .486 | .015 | 1.625 | < .001 |
| | Gender (male) | .230 | .186 | 1.259 | .225 |
| | Orientation[R] (horizontal) | -.187 | .080 | .830 | .021 |
| | Gender X SOA | -.084 | .022 | .919 | < .001 |

Coefficients, standard errors (SE), odds-ratios, and p-values are only reported for significant variables. The direction of significant effects in terms of conditions with higher likelyhood of responding correctly is given in brackets behind variables.

Models LeftRight and TopBottom are not presented, because there were no significant effects of variables LeftRight and TopBottom.

[R] random effects (as opposed to fixed effects)

[1] SOA is a scalar variable and the odds ratio was calculated per unit of change (i.e. per refresh frame of 8.33 ms)

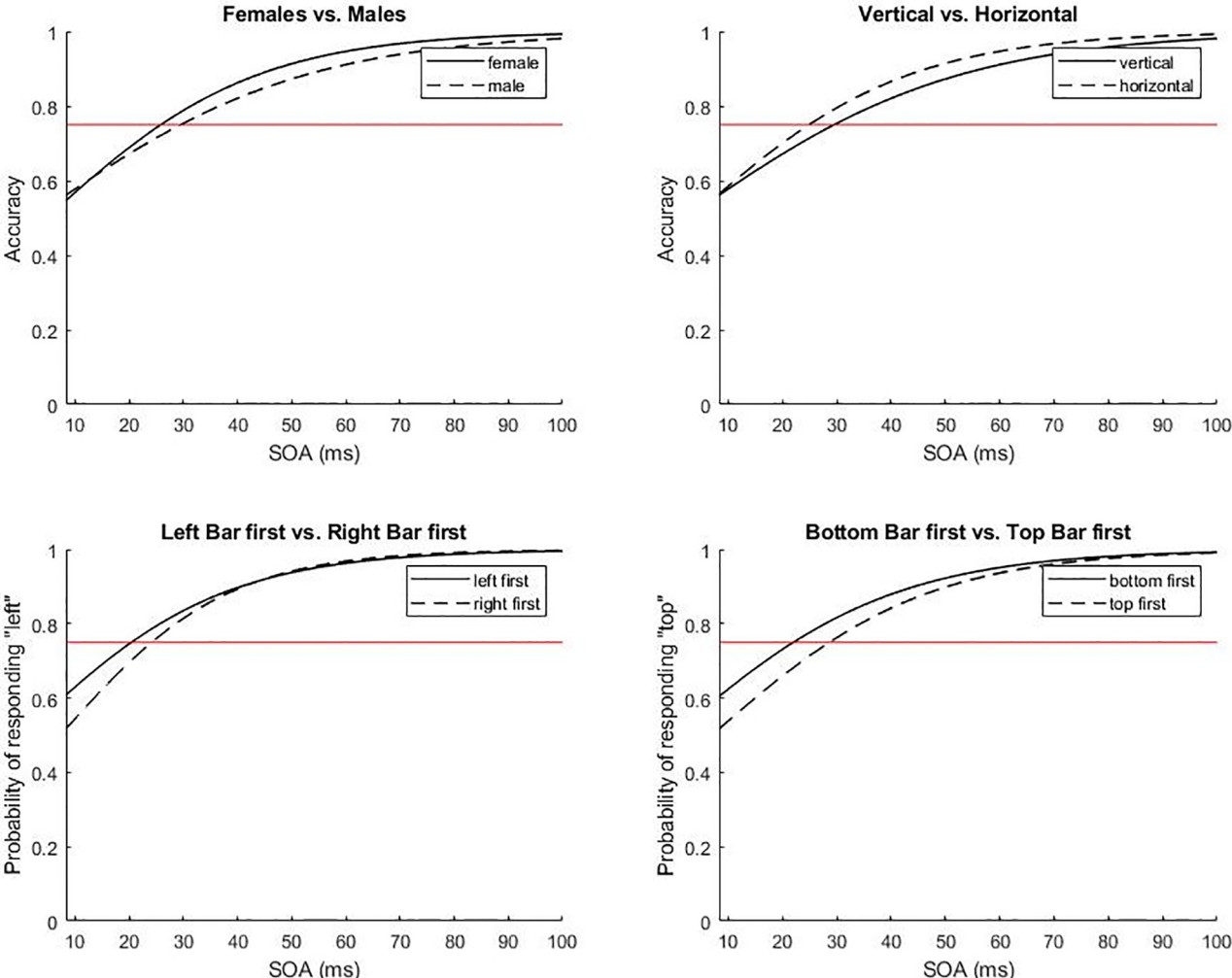

**Fig 3. The four plots represent illustration examples for factor combinations in the Orientation model (upper plots), the LeftRight model (lower left plot) and the TopBottom model (lower right plot).** Accuracy is defined as the probability of responding correctly over stimulus onset asynchronies (SOA) in the TOJ task for females (upper left; solid line) versus males (dashed line) and (upper right) vertical (solid line) versus horizontal bar depiction (dashed line). Bottom left graph: (lower left) probability of responding "left" over SOA for Left bar first (solid line) and Right bar first (dashed line). Bottom right graph: (lower right) probability of responding "top" over SOA for bottom bar first (solid line) versus top bar first (dashed line). The horizontal line indicates the temporal order threshold (75% correct responses).

participants at intermediate SOAs (see Fig 3), and between *SOA* and *Orientation*, showing that the accuracy differed across SOAs for horizontal versus vertical presentation, with higher accuracy for horizontal presentation.

The logistic regression analysis of the second model with *LeftRight* entered as a variable showed that although there was no significant main effect of the left bar being presented first versus the right bar being presented first, there was a significant interaction between *SOA* and *LeftRight*, indicating that the probability of responding "left" over initial SOAs was higher when the left bar was presented first than when the right one was presented first (see Table 4 and Fig 2). Similarly, in the third model with *TopBottom* entered as a variable, there was only a significant interaction between *SOA* and *TopBottom* (see Table 4 and Fig 2). As in the Orientation model, *SOA* again exerted a significant influence on *TOJ* in the LeftRight and the TopBottom models. Thresholds of TOJ can be seen in Table 5.

**Table 5. Thresholds of temporal order judgements.**

| Model | Variables | |
|---|---|---|
| Orientation | HTOJ | 21.5 (female) |
| | | 21.6 (male) |
| | VTOJ | 24.5 (female) |
| | | 25.2 (male) |

## 4. Discussion

Results of the current study show that several variables can influence event timing judgements (see Table 6 for an overview), suggesting that these judgements are to some extent fluid and that perceptual thresholds are dependent on a variety of individual, spatial and temporal factors. While previous research has shown that event timing can be modulated by such factors, the current study differs from previous experiments in two ways: (1) it uses an identical experimental setup where solely the two different task instructions (SJ and TOJ) differ and (2) it investigates a variety of factors at the same time. This methodological approach allows the assessment of whether the two event timing tasks, SJ and TOJ, rely on similar or different mechanisms, a question which has been under scientific debate. In the following, we discuss the present results, with respect to previous literature findings. We conclude that the dissociations between the SJ and TOJ tasks corroborate the idea that visual temporal event coding consists of several partially independent functions.

### 4.1 Individual factors

*Age*, *Sex* and *Daytime* were opportunistic variables included in the model to control their effect to the response. As the ratio of male to female participants (13:21) was not balanced, a limitation to be taken into account when interpreting the results.

**4.1.1 Age of participants.** We investigated whether the age of the participants modulated perception in a TOJ task and in a SJ task. The present results do not suggest that age has a modulatory influence on either of them. This is unexpected, and does not support our initial

**Table 6. Overview of significant factors.**

| TYPE OF FACTOR | VARIABLE | ROLE FOR SJ | ROLE FOR TOJ |
|---|---|---|---|
| Individual factors | Age | no | no |
| | Sex | **yes (resp. bias)** | **yes (sensitivity)** |
| Context I: Temporal Factors | SOA | **yes** | **yes** |
| | TrialN | **yes** | no |
| | Order_Exp | **yes** | no |
| | Order_Orientation | no | no |
| | Daytime (excl. night) | no | no |
| Context II: Spatial Factors | LeftRight | no | **yes** |
| | TopBottom | **yes** | **yes** |
| | Orientation | **yes (resp bias)** | **yes (sensitivity)** |

SJ = Simultaneity Judgements, TOJ = Temporal Order Judgements, SOA = Stimulus Onset Asynchrony, TrialN = trial number, Order_Exp = whether SJ or TOJ was conducted first, Order_Orientation = whether the horizontal or the vertical task version was conducted first, LeftRight = whether the left or right bar appeared first in any one trial in the horizontal task versions, TopBottom = whether the top or the bottom bar appeared first in any one trial in the vertical task versions, Orientation = whether stimuli were presented in a horizontal or vertical display.

hypothesis, since TOJ studies that directly compare samples of different ages with respect to their event timing abilities, while scarce, do confirm a modulatory effect in both auditory and visual TOJ tasks [55,56] and visual SJ [59]. However, it is likely that the present results are explained by the fact that the current sample was rather young (comparable to the young category in Fink and colleagues [29], for instance) and too homogeneous.

**4.1.2 Sex of participants.** We investigated whether the sex of the participants had a modulatory effect on visual temporal perception and have found this to indeed be the case for both the SJ and the TOJ tasks. In the present SJ task, male participants seemingly have lower thresholds than female participants. However, rather than reflecting more accurate timing, lower SJ thresholds in male participants seem to reflect a higher response bias towards the "asynchronous" answer option. For example, males were found to prefer it more often as females even when the stimuli were in fact synchronous (i.e., at 0 ms condition).

On the other hand, the findings also uncover a significant interaction between sex and SOA for the TOJ task, with accuracy increasing faster for females over SOA than for males. In this case, females seem to have a genuinely higher sensitivity in the TOJ task than males. This finding does not support our initial hypothesis and is in conflict with the previous literature which seems to suggest lower TOJ thresholds in men than in women either due to neuroanatomical or cognitive strategies differences [15,20,29,51–58]. However, given that the present results investigate such effects for the first time in the visual modality, it may be that Sex modulates performance in a TOJ task in the visual domain differently than it does in the auditory domain. Finally, the inclusion of the factor Sex in the study reveals a differential modulation of the SJ and TOJ, which might point to differing underlying mechanisms.

## 4.2 Contextual factors I: Temporal factors

**4.2.1 Daytime.** The time of day was not found to significantly influence perception in either the SJ or the TOJ tasks. This seemingly contradicts our initial hypothesis and findings by Lotze and colleagues [15] who assessed event timing performance over a period of 24 hours and found a modulatory effect for SJ but not TOJ. Certainly, the current study differs from the one by Lotze and colleagues in both stimulus properties (visual vs. auditory) and design (between-subject vs. within-subject). In addition, in Lotze and colleagues [15], the TOJ was in fact a *ternary* task. However, an even more obvious reason for the apparent conflict is that our participants were not tested in the evening or at nighttime, whereas Lotze and colleagues [15] report that SJ thresholds of auditory stimuli are most influenced during the night, with a maximum performance around midnight. This concurs with the psychophysical experiments in the visual system showing that threshold sensitivities follow a clear diurnal rhythm, which is 180˚ out of phase with those psychophysical functions which are mainly vigilance dependent [74].

**4.2.2 Stimulus onset asynchrony.** Confirming the initial hypothesis, SOA had a major influence on the performance in both SJ and TOJ tasks. Previous research has shown that humans tend to judge the onset of two visual stimuli to be simultaneous for stimulus onset asynchronies between 0 and about 60ms [9] and TOJ thresholds usually lie in the area of 20 and 60 ms, depending on physical stimulus properties [75]. As expected, our results showed that SOA modulates event timing perception in both SJ and TOJ tasks. It is important to note that comparison of the absolute thresholds is largely meaningless. First, because SJ and TOJ tasks produce different types of psychometric functions, there is no direct comparison between the two. Second, equally significantly, the criteria for both SJ and TOJ thresholds are matters of convention. While the SJ threshold is often 50% of a/synchronous responses and the TOJ threshold is most often set to 75% correct responses, other criteria have been used especially for the latter (e.g., [60] used 70.7% correct responses). Indeed, the standard thresholds (50%

and 75%) we used are no more "real" than any other criteria, and because the psychometric functions cannot be matched using the same criteria (e.g., 75% correct responses) they would not have the same meaning in both cases.

**4.2.3 Trial number.**   The investigation of the trial number (*TrialN*) factor brings up the issue of a potential training effect and the question of whether performance improves during the course of testing. Previously, several studies report empirical data directly supporting fast learning-induced improvements in TOJ tasks [49,50,60–65]. Few studies that have investigated training effects in SJ task have also shown positive learning effects for the onset and offsets of stimuli [49,50].

The present study is the first to investigate training effects in the visual domain by employing traditional SJ and TOJ tasks, as previous studies have focused on the auditory modality. Results show that a positive training effect occurs in the visual SJ task: Participants are almost twice as likely to respond "asynchronous" at the end of the testing). However, unlike the previous studies, we did not observe a training effect for the TOJ task.

We may ask why we found no training effect for the TOJ task, which has in the literature consistently been associated with improvements after training? One explanation appeals to differing methodologies, given that in previous studies the training lasted far longer than in our study. For instance, in the study by Mossbridge and colleagues [49], participants practiced the TOJ task one hour a day for six to eight days. This factor is unlikely to account for the whole effect, since their results show training effects also after a single exposure to the task (both for the test and for the control groups). For this reason, a more plausible alternative interpretation for why no training effects were found in our study is that there was little room to improve the performance. Hence, even if the training effect had occurred, it would have been too small to come up as statistically significant. This explanation is based on two claims. First, the TOJ threshold is thought to reflect the temporal resolution of a general, amodal time keeping mechanism. Second, the temporal threshold in the TOJ task (defined as 75% correct responses) is thought to be around 20 milliseconds. In our study, the threshold for males was 29.5 ms and for females 25.8 ms, both of which concur with the previously reported visual TOJ thresholds. Note that this is in stark contrast with the studies showing a training effect in the auditory modality, as there the initial threshold is in the range of 50 to 60 ms. The threshold lowers quickly and after an extensive training, it settles roughly at around 20 ms. Thus, given the two claims, it appears that our participants already reached the threshold and there was not much room for improvement. This explanation concurs with the fact that the training effect of the participants of Mossbridge and colleagues [49,50] decreased considerably the closer they were on their final personal threshold.

Unfortunately, our study does not allow us to answer the question why the thresholds for our participants were much closer to the threshold of trained individuals than that of untrained individuals. Nevertheless, together with the other studies, our results suggest that because of the changing results (thresholds) researchers should not put too much emphasis on the absolute SJ thresholds per se (especially if the differences are small). That is, the direct comparisons between the threshold, say between clinical population and controls, must check that one of the groups have not taken part on similar experiments before.

**4.2.4 Order of stimuli orientation.**   The question of whether performance was better depending on the order in which the horizontal or vertical stimuli orientation was experienced first, did not significantly modulate perception in either the SJ or the TOJ task. This could suggest that no learning effect manifest here, because learning takes (or does not take) place regardless of which task is done first.

**4.2.5 Order of task.**   The investigation of whether visual event timing performance is modulated by whether the SJ task or the TOJ task is conducted first brings up the issue of

generalization patterns across SJ and TOJ tasks. This issue has been studied earlier by Mossbridge and colleagues [49,50] with regard to the onsets and offsets of auditory stimuli. While they found no evidence for any generalization of the effects of training across SJ and TOJ tasks for the onsets of auditory stimuli, they reported that learning in SJ task involving the offsets of auditory stimuli generalized to the TOJ task.

In our study, participants were more likely to respond "asynchronous" in the SJ task, when the SJ task was performed before the TOJ task compared to the other way around. This suggests that a sharpening of perception or a temporal calibration takes place, if a person is trained by judging TOJ, which subsequently increases SJ performance. This generalization is likely to occur via an automatic mechanism, like implicit or perceptual learning. On the other hand, unlike in Mossbridge and colleagues [50], the accuracy in the TOJ task was not modulated by the order in which the two experiments were performed. Therefore, the results of our study disagree with those of Mossbridge and colleagues [50]. One possible explanation for why we did not find a generalization effect from the SJ task to the TOJ task is the same as above: If the threshold of visual TOJ is already close to the lowest thresholds (as it was for our participants), there is no room for a generalization pattern to manifest in this case too. Another possible explanation for this discrepancy in findings is the nature of stimuli used in the studies. Not only did our stimuli differ from theirs with respect to modality, they also differed with regard to the rise time: The rise and fall time of their auditory stimuli was 10 ms, whereas the rise time of our stimuli was 41.67ms. Ours was to prevent the sense modality specific interaction that might distort the results, i.e., perception of apparent motion. Given the suggestion that auditory modality has a rise-time-sensitive mechanism, independent of sound intensity changes, it is possible that this has influenced their results [76]. This would also explain why there was no generalization effect when the time marker was the onset of stimuli [49] but there was one when the time marker was the offset of stimuli [50].

## 4.3 Contextual factors II: Spatial factors

We investigated the role of spatial factors in both SJ and TOJ tasks: (1) *LeftRight* (i.e. whether the left or right bar appeared first in the horizontal task versions), (2) *TopBottom* (i.e. whether the top or the bottom bar appeared first in any one trial in the vertical task versions) and (3) *Orientation* (i.e. whether the stimuli were presented in horizontal or vertical orientation). These factors all had an effect on both SJ and TOJ except for the *LeftRight* factor, which did not modulate SJ.

**4.3.1 Order of stimulus location (LeftRight and TopBottom).** The LeftRight factor was concerned with whether the left or the right bar appeared first in the horizontal task versions and how this affected event timing perception. Results support our initial hypothesis and indicate a difference between SJ and TOJ tasks in that only in the latter there was a significant interaction between the LeftRight factor with SOA. In particular, the probability of responding "left" over the very small SOAs was higher when the left bar was presented first than when the right one was presented first. This confirms previous findings in visual TOJ. For instance, Sekuler, Tynan and Levinson, [66] have found a general bias, whereby "*Observers tend to report that the left target flashed before the right target, regardless of the actual order in which they were presented*" (p. 180). The authors conclude that this tendency could be due to a left–right scanning mechanism such as is assumed to occur in reading, which would make this an attentional effect. The prior entry hypothesis states that attention accelerates sensory processing, shortening the time to perception [77–79]. However, Schneider and Bavelier [42] offer alternative explanations like response biases or other changes in the decision criteria that might be interpreted as prior entry effects. In the present study it is difficult to distinguish between

attentional effects upon sensory mechanisms and those upon cognitive mechanisms. But given that the two curves do not coincide at the first SOA condition of 8.33 ms, we might infer that the finding is due to a cognitive bias rather than an actual sensitivity difference (see Fig 2, lower left). The lack of an effect on SJ could be explained by the fact that, attentional effects on SJ have been mostly associated with exogenous cues (i.e. sensory changes near the target locations, [42]).

The TopBottom factor was concerned with whether the top or the bottom bar appeared first in any one trial in the vertical task versions. In the SJ task, analyses show that when the bottom bar appeared first, response bias increased and participants were more liberal at judging stimuli as non-simultaneous. For the TOJ task, this factor also modulated performance as an interaction with SOA so that with increasing SOA, the response bias of responding "top" for bottom bar first decreased. As this has not been investigated hitherto, we speculate that this may be due to internalized Newtonian mechanics–because of the internalized laws of gravity, top-bottom and bottom-top are not perceived equivalently [80]. Therefore, if top precedes bottom, a stimulus may be perceived as "falling down" due to experience with gravity and therefore there is more binding in such a case compared to if something is presented from bottom to top.

**4.3.2 Orientation.** The Orientation factor was concerned with whether performance in the horizontal orientation was better than in the vertical orientation of the SJ and TOJ tasks. In the present study, orientation was found to have an effect in both tasks, namely a superior TOJ performance and an increased response bias in the SJ task for the horizontal compared to the vertical stimulus presentation. This highlights an interesting difference between the two SJ and TOJ tasks. In the SJ task participants were more liberal in responding "asynchronous" for the horizontal bar depiction, revealing a response bias. To our knowledge, only one other study has investigated the orientation factor in an SJ task and did not find any significant differences between target orientations [44]. However, compared to the present study, the target stimuli in the study by Carver and Brown [44] were preceded by correct or incorrect attentional cues. Thus, one possibility could be that such cues, may cancel the response bias (or render it comparable). Furthermore, participants in the present study tended to be more precise in temporal order judgements when judging horizontal rather than vertical stimuli. We can speculate that the current results may be related to the fact that most of the time we manipulate things in the horizontal rather than the vertical plane (e.g. moving things from one place to another, our own movements, including eye-movements, take place mostly horizontally). Therefore, we speculate that this may be a consequence of perceptual learning. Alternatively, it might also be of disadvantage to bind events together, except for top to bottom stimuli, as this happens rarely (except when one single object falls). The fact that *LeftRight*, *RightLeft* and *BottomTop* all have roughly similar TOJ thresholds seems to be supporting this.

## 4.4. Similarities and differences of SJ and TOJ and implications for underlying mechanisms

Despite finding that most of the included variables modulated event timing perception, some interesting dissociations between SJ and TOJ emerge, demonstrating that in some cases said variables only affect one type of judgement but not the other. Furthermore, other variables seem to modulate event timing perception for both types of judgement, yet they do so in different ways. These findings corroborate the idea that visual temporal event coding consists of several partially independent functions [33,41,43,81,82].

Sex as well the Orientation of the stimuli were found to modulate both SJ and TOJ performance, however, in the case of the first, they revealed a response bias in favour of the male

participants, whereas in the case of the latter, a true sensitivity in favour of the female participants. Given that SJ requires participants to judge the simultaneity of two events, this may bias them toward assuming that the stimuli should go together (i.e., they may have explicitly tried to bind them) in accordance with the "unity assumption", whereby, an observer assumes that two different sensory signals refer to the same multisensory event [83,84]. Whether the left or the right bar appeared first in the horizontal task versions (*LeftRight*) seemed to affect the TOJ task only. In turn, the training and perceptual learning effects revealed by the number of trails (*TrailN*) and by the order in which the two tasks were performed (*Order_Exp*) modulated only SJ.

Our findings support previous research that suggests that the two tasks are supported by partially different mechanisms. Van Eijk and colleagues [43] summarized results from studies aimed at estimating the "point of subjective simultaneity" (PSS) derived from TOJ and SJ tasks. They showed that the two tasks generally provide discrepant estimates. Importantly, PSS obtained using SJ and TOJ tasks are thought to reflect the same phenomena and they can be compared. This is unlike our use of two separate psychometric functions, which is widely used, but makes a direct comparison meaningless. The authors also carried out a within-subjects study involving three types of tasks (SJ, TOJ and a ternary task), with the goal to investigate the influence of experimental method and stimulus type on estimates of audio–visual timing perception. Their results confirmed that thresholds from the SJ and ternary tasks were highly correlated, whereas PSS estimates from TOJ tasks were uncorrelated with those tasks. Their results also revealed that PSS estimates from SJ and ternary tasks did not differ significantly, whereas PSS estimates from TOJ tasks were significantly lower than those from SJ or ternary tasks. The reason for the discrepant PSS estimates in SJ and TOJ tasks is unclear, but these discrepancies have prompted the view that SJ and TOJ tasks measure distinct processes and involve different response biases [33,41,43,81,82,85,86]. It is important to note that Linares and Holcombe [81] suggest that each judgment is subject to different processes that bias perceptual latency in different ways: TOJ might be affected by sensory interactions, a bias associated with the method of single stimuli and an order difficulty bias. On the other hand, SJ might be affected by sensory interactions and an asymmetrical criterion bias. García-Pérez and Alcalá-Quintana [41] present a model including sensory and decisional parameters that places the SJ and TOJ tasks in a common framework which allows studying their implications on observed performance. They suggest that the TOJ tasks implies specific decisional components that explain the discrepancy of results obtained with TOJ and SJ tasks. This view is corroborated by Miyazaki and colleagues [33] who find in their neuroimaging study that TOJ-specific activity was indeed observed in multiple regions that overlap with the general temporal prediction network for perception and motor systems, whereas SJ-specific activation, on the other hand, was only observed in the posterior insula.

Vatakis and colleagues [82] wanted to examine whether the measures of participants' temporal discrimination performance derived from the TOJ and SJ tasks was related (as one might expect if one thinks that they are measuring the same underlying psychological phenomena). Therefore, they conducted a correlation analysis between the TOJ and SJ data on a participant-by-participant basis and found no correlation between the performance in the TOJ and SJ tasks, thus providing support for the claim that the two tasks may actually measure different aspects of temporal perception. In their study, however, participants had to additionally monitor an asynchronous (as opposed to a synchronous) background speech stream and the stimuli were bimodal (flash and burst of noise). In fact, all these studies (with the exception of [86], which was a prior entry study) have used multimodal stimuli. Given that inter-modal stimulation (i.e. concurrent stimulation in different sensory modalities) leads to the highest thresholds [2,14,15]. The present study therefore provides evidence using a unimodal paradigm to

confirm the view that SJ and TOJ are two separate functions. Furthermore, the spatial and learning effects uncovered in the present study with the current design have interesting implications: SJ seems somehow subordinary to TOJ, as it is not able to influence TOJ, but TOJ is able to influence SJ. This is in line with the view that the perception of nonsimultaneity is a necessary, but not a sufficient condition for the perception of TOJ [5].

The results of the current study present a set of factors influencing processing of stimulus timing in healthy individuals, which could serve as baseline for studies of time processing in psychopathologies, and we suggest these factors to be taken into account to investigate potentially different effects in different populations. Further factors that should be taken into account by future studies relate to a systematic variation of spatial distances between stimuli (which were fixed in the current study), which might influence perceptual resolution. In addition, individual trait factors such as autistic traits or impulsivity traits might arguably influence perception and response criteria and should be studied systematically in relation to event timing in human vision.

## 5. Conclusion

The current study uses a unique experimental setup and stimuli to test the stability of unimodal, visual event timing perception. Compared to previous research, the advantages of the current methodology lie with the implementation of the same unimodal visual stimuli for both SJ and TOJ tasks and the investigation of many individual and contextual (temporal and spatial) factors. Given that stimuli and setup were the same across tasks, sensory aspects are likely to have been invariant across tasks, with differences across them only in criteria or biases. Results show that several explanatory variables modulate SJ and TOJ suggesting that these judgements about the timing of events are to some extent fluid and the thresholds are dependent on various factors, resulting in variability. Moreover, results suggest that SJ and TOJ may be modulated differently by the same variables. This provides further evidence that unimodal SJ and TOJ are partially independent functions. This cannot be concluded otherwise on the basis of thresholds alone, because the tasks demands are too different.

## Supporting information

**S1 File.**
(R)

## Acknowledgments

We would like to thank the participants for volunteering their time and thoughts. This work was funded under the Volkswagen Foundation grant I/82 894 awarded to VA, VN and CFW. Dr. Daniel Lunn passed away before the submission of the final version of this manuscript. Christine Falter-Wagner accepts responsibility for the integrity and validity of the data collected and analyzed.

## Author Contributions

**Conceptualization:** Valtteri Arstila, Valdas Noreika, Christine M. Falter-Wagner.

**Data curation:** Valtteri Arstila, Henri Pesonen, Daniel Lunn, Valdas Noreika.

**Formal analysis:** Alexandra L. Georgescu, Henri Pesonen, Daniel Lunn, Valdas Noreika, Christine M. Falter-Wagner.

**Funding acquisition:** Valtteri Arstila, Valdas Noreika, Christine M. Falter-Wagner.

**Investigation:** Valtteri Arstila.

**Methodology:** Valtteri Arstila, Alexandra L. Georgescu, Henri Pesonen, Daniel Lunn, Valdas Noreika, Christine M. Falter-Wagner.

**Project administration:** Valtteri Arstila, Valdas Noreika, Christine M. Falter-Wagner.

**Resources:** Valtteri Arstila, Alexandra L. Georgescu, Valdas Noreika, Christine M. Falter-Wagner.

**Supervision:** Daniel Lunn.

**Validation:** Valtteri Arstila, Alexandra L. Georgescu, Henri Pesonen, Valdas Noreika.

**Visualization:** Valtteri Arstila, Alexandra L. Georgescu.

**Writing – original draft:** Valtteri Arstila, Alexandra L. Georgescu, Valdas Noreika.

**Writing – review & editing:** Valtteri Arstila, Alexandra L. Georgescu, Henri Pesonen, Valdas Noreika, Christine M. Falter-Wagner.

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
