## [Decision Letter · Decision Letter 0]

16 Mar 2020

PONE-D-19-31687

Event Timing in Human Vision: Modulating Factors and Independent Functions

PLOS ONE

Dear Dr. Falter-Wagner,

Thank you for submitting your manuscript to PLOS ONE. After careful consideration, we feel that it has merit but does not fully meet PLOS ONE’s publication criteria as it currently stands. Therefore, we invite you to submit a revised version of the manuscript that addresses the points raised during the review process.

We would appreciate receiving your revised manuscript by Apr 30 2020 11:59PM. To enhance the reproducibility of your results, we recommend that if applicable you deposit your laboratory protocols in protocols.io, where a protocol can be assigned its own identifier (DOI) such that it can be cited independently in the future. For instructions see: http://journals.plos.org/plosone/s/submission-guidelines#loc-laboratory-protocols

We look forward to receiving your revised manuscript.

Kind regards,

Robin Baurès, Ph.D.

Academic Editor

PLOS ONE

Journal Requirements:

Additional Editor Comments (if provided):

Dear Dr. Falter-Wagner

I have now received two reviews regarding your manuscript. As you will see, the two reviewers have a different opinion on your manuscript. The strongest concerns come from R1, who points out major concerns and would suggest refusing your manuscript. The comments seem quite justified to me, however, I am not sure this comments could not be addressed. In particular the main comment 1 regarding the statistics could be addressed by doing the stats as advised here (or justifying if possible why the stats were correctly performed). Regarding point 3, I also agree that the very low number of males tested seems a problem to me, in particular if you aim at comparing the males performances to the females and that the two numbers are very unbalanced. I would therefore suggest testing an additional 10 male participants.

In addition to these specific points, would you choose to resubmit your manuscript, please take care to answer to all the comments I am not specifically listing here.

I hope you will be able to answer each of the comments that have been formulated by the reviewers, and do thank you for considering Plos One for publishing your work.

Best,

Robin Baurès

Journal Requirements:

4. Please include your tables as part of your main manuscript and remove the individual files. Please note that supplementary tables (should remain/ be uploaded) as separate "supporting information" files

Reviewers' comments:

Reviewer's Responses to Questions

**Comments to the Author**

1. Is the manuscript technically sound, and do the data support the conclusions?

Reviewer #1: No

Reviewer #2: Partly

2. Has the statistical analysis been performed appropriately and rigorously? 

Reviewer #1: No

Reviewer #2: Yes

3. Have the authors made all data underlying the findings in their manuscript fully available?

Reviewer #1: No

Reviewer #2: No

4. Is the manuscript presented in an intelligible fashion and written in standard English?

Reviewer #1: Yes

Reviewer #2: Yes

5. Review Comments to the Author

Reviewer #1: I am really sorry, but I suggest rejection. The manuscript has a nice structure and useful theoretical summaries in the first part of its introduction. However such major changes would be needed that could easily result in an entirely different work.

Major points:

1. The authors should report their results in accordance with APA or similar standards indicating degrees of freedom. It seems that the binary response of a single trial was used as independent variable, so each trial of a participant was treated as an individual case (N = 260x30x2?). This procedure is not correct because the elements of the sample are not independent. With a sample size of around 156000 it is not surprising to gain many significant results, however, those results do not necessarily indicate real effects characterizing the population.

2. Albeit the introduction proposes interesting questions, like how the two tasks are different (whether simultaneity perception is a prerequisite of temporal judgement), the authors can only conclude that the two tasks measure partially different functions. That is not surprising at all (Binder, 2015; Linares & Holcome, 2014; Love et al, 2013; Matthews et al, 2016; Miyazaki et al, 2016; Van Eijk et al, 2008, 2010; Vatakis et al, 2008) given that even different versions of each task can measure different underlying constructs (Fostick & Babkoff, 2011, 2013, 2014).

3. The strongest claim of the study is that it measures multiple variables with comparable TOJ and SJ tasks, but the sample size is too small. Ten men do not represent the male population. This fact and the method problems mentioned in point 1 can result in unexpected findings like females having higher sensitivity in TOJ. Furthermore, the authors admittedly used a sample that was quite homogeneous regarding the individual variables (mainly young females) while both variables sex and age were part of their experimental design.

Reviewer #2: The current study directly compared various aspects of judgments of simultaneity (SJ) and of temporal order (TOJ) by using multilevel binary regression modelling. The authors investigated modulatory effects of potential explanatory variables on event timing perception: (1) Individual factors (sex and age), (2) temporal factors (SOA, trial number, order of experiment, order of stimuli orientation, time of day) and (3) spatial factors (left or right stimulus first, top or bottom stimulus first, horizontal vs. vertical orientation), and the authors found the modulatory effects in parts of them. Then they concluded that SJ and TOJ are partially independent functions, because they are modulated differently by individual and contextual variables.

In contrast to traditional analysis using several metrics like JND, the present approach looks like very unique, and informative in considering various factors like listed above. However, the manuscript includes several points to be clarified and corrected as listed below.

Major points.

First of all, explanation about multilevel binary regression modelling is definitively lacked. The authors need to describe the method and equations so that readers can understand easily. I guess readers of PLoS one is widely distributed in all fields of science. In addition, I’m afraid formats of references are wrong. Please add and reconsider the explanations and references.

Second, although the authors considered several factors as listed above, there are still several factors to be considered. For example,

1. How was a distance between 2 stimuli? To my experiences, temporal resolutions seem to be dramatically changed by the distances during visual TOJ.

2. How was a brightness of the stimuli? Brightness also affects the judgment.

3. How was characteristics of the participants? Several characteristics related to autism, ADHD and so on also affect the result.

I understand it is very difficult to control the factors. But I think discussion about these factors are needed as limitations of the study.

Minor points

1. Description about the participants were unclear. What is ‘remaining 10 males’? In my understandings, there are 10 male participants and 20 female participants. In addition, one participant was excluded. Please indicate the criteria of the exclusion more precisely.

2. Ratio of male and female was not same. It is a little problematic for analysis of sex difference. Please control the ratio, or please add the discussion as a limitation.

3. Were the task procedures identically same to the previous studies? Otherwise, the authors need to describe the task set-up, apparatus and procedures more precisely.

4. Author declared that all data are fully available without restriction and the data will be shared publicly on OSF. At this time, response data of the participants are not available. Thus, I choose “No” at the review question-3.

6. PLOS authors have the option to publish the peer review history of their article (what does this mean?). If published, this will include your full peer review and any attached files.

Reviewer #1: No

Reviewer #2: No

---

## [Author Response · Author response to Decision Letter 0]

3 May 2020

Reviewer #1: I am really sorry, but I suggest rejection. The manuscript has a nice structure and useful theoretical summaries in the first part of its introduction. However such major changes would be needed that could easily result in an entirely different work.

Major points:

1. The authors should report their results in accordance with APA or similar standards indicating degrees of freedom. It seems that the binary response of a single trial was used as independent variable, so each trial of a participant was treated as an individual case (N = 260x30x2?). This procedure is not correct because the elements of the sample are not independent. With a sample size of around 156000 it is not surprising to gain many significant results, however, those results do not necessarily indicate real effects characterizing the population.

Response:

The results in this article are reported in accordance with APA and we would like to politely point the reviewer to the APA primer on multilevel modelling:

https://www.apa.org/science/about/psa/2017/01/multilevel-modelling

Indeed, the binary response was used as independent variable and it is correct that the elements of the sample are not independent. The relevant section of the APA primer says:

“The issue with this approach is that we repeatedly sampled from the same individual, and this violates the OLS assumption that observations are independent from each other. Instead, observations from the same individual are likely to be “clustered,” and a MLM adds some extra parameters that control for this clustering. In particular, MLMs add structure to the error term from OLS. Instead of one general random-effect that captures how each observation deviates from the predicted fixed-effects, there will be multiple random-effects that capture how observations deviate within a cluster, and how each cluster deviates from the overall group. We estimate the variability for each random-effect and use that to control for the variance when estimating the significance our fixed-effects. Thus, we can model our data at the observation level (micro-level) and at the cluster level (macro-level). This combination of different “levels” of analysis gives rise to the term multi-level modeling.”

Specifically, in our current study all our test situations follow a repeated measurements design so that measurements are nested within subjects and not independent from each other. Multi-level models allow us to fit logistic regression models at subject-level by allowing the intercept-term and the coefficient for the test type-covariate to vary for each individual (Gelman & Hill, 2006). This in effect have allowed to model the clustering of the responses of each individual. We have added the details of our logistic multi-level models in the Statistical Analysis-section to make this more evident. 

2. Albeit the introduction proposes interesting questions, like how the two tasks are different (whether simultaneity perception is a prerequisite of temporal judgement), the authors can only conclude that the two tasks measure partially different functions. That is not surprising at all (Binder, 2015; Linares & Holcome, 2014; Love et al, 2013; Matthews et al, 2016; Miyazaki et al, 2016; Van Eijk et al, 2008, 2010; Vatakis et al, 2008) given that even different versions of each task can measure different underlying constructs (Fostick & Babkoff, 2011, 2013, 2014).

Response:

Yes, we agree with the reviewer that it can be concluded from the results that SJ and TOJ are based on (partially) different functions. Whether that may be surprising or not, the current study provides analyses of a series of factors modulating human event timing and by showing different modulations confirming independent functions of SJ and TOJ as well as showing that fixed thresholds, as frequently reported in the literature so far, are in fact relative in consideration of this modulation. Please also note that the novelty of results is not a criterion in the PLOS One review process: “We evaluate submitted manuscripts on the basis of methodological rigor and high ethical standards, regardless of perceived novelty”. https://journals.plos.org/plosone/s/journal-information

However, given the interesting addition that even different task versions of the same task can lead to different results, we now added the following sentence in the introduction:

“However, huge variability in TOJ thresholds has also been observed, depending on the physical properties of the stimuli (for a review, see Wittmann, 2011a) and even different task version (Fostick, Ben-Artzi, & Babkoff, 2011; Fostick & Babkoff, 2013; Fostick, Babkoff, & Zukerman 2014).” 

3. The strongest claim of the study is that it measures multiple variables with comparable TOJ and SJ tasks, but the sample size is too small. Ten men do not represent the male population. This fact and the method problems mentioned in point 1 can result in unexpected findings like females having higher sensitivity in TOJ. Furthermore, the authors admittedly used a sample that was quite homogeneous regarding the individual variables (mainly young females) while both variables sex and age were part of their experimental design.

Response:

We correct that we had tested 13 males and 21 females, as correctly stated in the current version of the manuscript. Nevertheless, the factor Sex was not part of our experimental design. This was already acknowledged in the manuscript: “Whereas the inclusion of the majority of these factors was planned, Age, Sex and Daytime were opportunistic variables, the results of which must be interpreted with caution.” To make this clearer, we now repeat in the discussion section: “Age, Sex and Daytime were opportunistic variables, the results of which must be interpreted with caution.”

Reviewer #2: The current study directly compared various aspects of judgments of simultaneity (SJ) and of temporal order (TOJ) by using multilevel binary regression modelling. The authors investigated modulatory effects of potential explanatory variables on event timing perception: (1) Individual factors (sex and age), (2) temporal factors (SOA, trial number, order of experiment, order of stimuli orientation, time of day) and (3) spatial factors (left or right stimulus first, top or bottom stimulus first, horizontal vs. vertical orientation), and the authors found the modulatory effects in parts of them. Then they concluded that SJ and TOJ are partially independent functions, because they are modulated differently by individual and contextual variables.

In contrast to traditional analysis using several metrics like JND, the present approach looks like very unique, and informative in considering various factors like listed above. However, the manuscript includes several points to be clarified and corrected as listed below.

Response: 

Many thanks to the reviewer for acknowledging the value of the chosen approach.

Major points.

First of all, explanation about multilevel binary regression modelling is definitively lacked. The authors need to describe the method and equations so that readers can understand easily. I guess readers of PLoS one is widely distributed in all fields of science. In addition, I’m afraid formats of references are wrong. Please add and reconsider the explanations and references.

Response:

Thanks for pointing these issues out. Please refer to our response no. 1 concerning the statistical approach. We have included the equations in the text and R script used to run the analyses as supplementary material. Concerning the references, we corrected the format in the current version of the manuscript. 

Second, although the authors considered several factors as listed above, there are still several factors to be considered. For example,

1. How was a distance between 2 stimuli? To my experiences, temporal resolutions seem to be dramatically changed by the distances during visual TOJ.

Response: 

The distance was “11.1 degrees of visual angle between their centres” as stated in the Methods section. 

2. How was a brightness of the stimuli? Brightness also affects the judgment.

As stated in the Methods section: “The bar stimuli were faded in incrementally within 5 frames (8.33ms per frame) from 5.2 – 53.7 lux.”

3. How was characteristics of the participants? Several characteristics related to autism, ADHD and so on also affect the result.

I understand it is very difficult to control the factors. But I think discussion about these factors are needed as limitations of the study.

Response:

All participants were healthy and had no psychiatric diagnosis. The aim of the study was not to investigate (potentially) pathological processing of timing but deliver a set of factors influencing processing of stimulus timing in healthy individuals, which could serve as baseline for studies of time processing in psychopathologies. We added this sentence in the discussion: “The results of the current study present a set of factors influencing processing of stimulus timing in healthy individuals, which could serve as baseline for studies of time processing in psychopathologies and we suggest these factors to be taken into account to investigate potentially different effects in different populations.” 

Minor points

1. Description about the participants were unclear. What is ‘remaining 10 males’? In my understandings, there are 10 male participants and 20 female participants. In addition, one participant was excluded. Please indicate the criteria of the exclusion more precisely.

Response: One participants was excluded before any further analysis, as he showed random response behavior. We added in the methods section:

Data of one participant was not included in the analysis due to random response behaviour (performance up to 4 SD below the group mean).

2. Ratio of male and female was not same. It is a little problematic for analysis of sex difference. Please control the ratio, or please add the discussion as a limitation.

Response:

We have now added the ratio as a limitation in the discussion: “(…) the ratio of male to female participants (13:21) was not balanced, a limitation to be taken into account when interpreting the results.”

3. Were the task procedures identically same to the previous studies? Otherwise, the authors need to describe the task set-up, apparatus and procedures more precisely.

Response: 

The horizontal simultaneity judgment task was an exact replication. The text now makes this clearer and states:

“The HSJ task was exactly replicated from our previous study (Falter, Elliott, & Bailey, 2012) and the VSJ version was adapted to vertical stimulus presentation. The HTOJ and VTOJ were exactly the same tasks only with different instruction (i.e. asking participants to attend to the temporal order of stimuli instead of their simultaneity).”

4. Author declared that all data are fully available without restriction and the data will be shared publicly on OSF. At this time, response data of the participants are not available. Thus, I choose “No” at the review question-3.

Response:

We have now uploaded the data on OSF (DOI 10.17605/OSF.IO/3KHDG).

---

## [Decision Letter · Decision Letter 1]

23 Jul 2020

PONE-D-19-31687R1

Event Timing in Human Vision: Modulating Factors and Independent Functions

PLOS ONE

Dear Dr. Falter-Wagner,

Thank you for submitting your manuscript to PLOS ONE. After careful consideration, we feel that it has merit but does not fully meet PLOS ONE’s publication criteria as it currently stands. Therefore, we invite you to submit a revised version of the manuscript that addresses the points raised during the review process.

I would like first to apologize for the time it took me to send you this decision. It appears that the most critical reviewer (R1) did not wanted to read &nd comment your revised version of the manuscript, stating that it should have been rejected without a possible revision. We therefore looked for a new reviewer who would have been reading your manuscript with a special emphasis on the statistics. It happened that it is a hard task, and we only got refusals from a large number of possible reviewer. We have therefore decided to move forward without this reading, based on my own judgment of your answer to R1. It eventually only remains the R2 minor points that I would like you to address to definitely accept the submission.

We look forward to receiving your revised manuscript.

Kind regards,

Robin Baurès, Ph.D.

Academic Editor

PLOS ONE

Reviewers' comments:

Reviewer's Responses to Questions

**Comments to the Author**

1. If the authors have adequately addressed your comments raised in a previous round of review and you feel that this manuscript is now acceptable for publication, you may indicate that here to bypass the “Comments to the Author” section, enter your conflict of interest statement in the “Confidential to Editor” section, and submit your "Accept" recommendation.

Reviewer #2: (No Response)

2. Is the manuscript technically sound, and do the data support the conclusions?

Reviewer #2: Yes

3. Has the statistical analysis been performed appropriately and rigorously? 

Reviewer #2: Yes

4. Have the authors made all data underlying the findings in their manuscript fully available?

Reviewer #2: Yes

5. Is the manuscript presented in an intelligible fashion and written in standard English?

Reviewer #2: Yes

6. Review Comments to the Author

Reviewer #2: I think the revised manuscript is generally improved with regards to my pointed out. However, I still have several minor questions and suggestions to the authors.

> The distance was “11.1 degrees of visual angle between their centres” as stated in the Methods section.

In my previous question, I meant a distance between 2 stimuli could be crucial factor that affects the temporal resolution, although I understand this factor was fixed in the present study. I suggest that the authors add discussions about this as a possible spatial factor.

> As stated in the Methods section: “The bar stimuli were faded in incrementally within 5 frames (8.33ms per frame) from 5.2 – 53.7 lux.”

My question was that then the brightness of the bar was kept at 53.7 lux till end of the trial?

> The aim of the study was not to investigate (potentially) pathological processing of timing but deliver a set of factors influencing processing of stimulus timing in healthy individuals, which could serve as baseline for studies of time processing in psychopathologies.

I basically agree the reply and additional description of the authors. But, I’m still thinking that autistic traits is known to be distributed in healthy populations widely (i.e., autism spectrum), and then it could affect the results. I suggest adding short discussion about this as a possibilities individual factor.

7. PLOS authors have the option to publish the peer review history of their article (what does this mean?). If published, this will include your full peer review and any attached files.

Reviewer #2: No

---

## [Author Response · Author response to Decision Letter 1]

27 Jul 2020

Reviewer #2: I think the revised manuscript is generally improved with regards to my pointed out. However, I still have several minor questions and suggestions to the authors.

1. The distance was “11.1 degrees of visual angle between their centres” as stated in the Methods section.

In my previous question, I meant a distance between 2 stimuli could be crucial factor that affects the temporal resolution, although I understand this factor was fixed in the present study. I suggest that the authors add discussions about this as a possible spatial factor.

Response:

We agree with the reviewer and have added to the Discussion section the following discussion: “Further factors that should be taken into account by future studies relate to a systematic variation of spatial distances between stimuli (which were fixed in the current study), which might influence perceptual resolution.”

2. As stated in the Methods section: “The bar stimuli were faded in incrementally within 5 frames (8.33ms per frame) from 5.2 – 53.7 lux.”

My question was that then the brightness of the bar was kept at 53.7 lux till end of the trial?

Response:

We have added to the Methods section: “(and kept at that brightness until the response)”

3. The aim of the study was not to investigate (potentially) pathological processing of timing but deliver a set of factors influencing processing of stimulus timing in healthy individuals, which could serve as baseline for studies of time processing in psychopathologies.

I basically agree the reply and additional description of the authors. But, I’m still thinking that autistic traits is known to be distributed in healthy populations widely (i.e., autism spectrum), and then it could affect the results. I suggest adding short discussion about this as a possibilities individual factor.

Response:

We have added to the Discussion section: “In addition, individual trait factors such as autistic traits or impulsivity traits might arguably influence perception and response criteria and should be studied systematically in relation to event timing in human vision.”

---

## [Editor Report · Decision Letter 2]

30 Jul 2020

Event Timing in Human Vision: Modulating Factors and Independent Functions

PONE-D-19-31687R2

Dear Dr. Falter-Wagner,

We’re pleased to inform you that your manuscript has been judged scientifically suitable for publication and will be formally accepted for publication once it meets all outstanding technical requirements.

Kind regards,

Robin Baurès, Ph.D.

Academic Editor

PLOS ONE
---

## [Editor Report · Acceptance letter]

5 Aug 2020

PONE-D-19-31687R2 

Event Timing in Human Vision: Modulating Factors and Independent Functions 

Dear Dr. Falter-Wagner:

I'm pleased to inform you that your manuscript has been deemed suitable for publication in PLOS ONE. Congratulations! Your manuscript is now with our production department. 

Kind regards, 

on behalf of

Dr. Robin Baurès 

Academic Editor

PLOS ONE